# Copper Overload Increased Rat Striatal Levels of Both Dopamine and Its Main Metabolite Homovanillic Acid in Extracellular Fluid

**DOI:** 10.3390/ijms25158309

**Published:** 2024-07-30

**Authors:** Antón Cruces-Sande, Pablo Garrido-Gil, Germán Sierra-Paredes, Néstor Vázquez-Agra, Álvaro Hermida-Ameijeiras, Antonio Pose-Reino, Estefanía Méndez-Álvarez, Ramón Soto-Otero

**Affiliations:** 1Laboratory of Neurochemistry, Department of Biochemistry and Molecular Biology, Faculty of Medicine, University of Santiago de Compostela, 15782 Santiago de Compostela, Spain; german.sierra.paredes@usc.es (G.S.-P.); estefania.mendez@usc.es (E.M.-Á.); ramon.soto@usc.es (R.S.-O.); 2Health Research Institute of Santiago de Compostela (IDIS), 15706 Santiago de Compostela, Spain; pablo.garrido@usc.es (P.G.-G.); alvaro.hermida@usc.es (Á.H.-A.); antonio.pose@usc.es (A.P.-R.); 3Laboratory of Cell and Molecular Neurobiology of Parkinson’s Disease, Department of Morphological Sciences, Faculty of Medicine, Research Center for Molecular Medicine and Chronic Diseases (CIMUS), University of Santiago de Compostela, 15782 Santiago de Compostela, Spain; 4Networking Research Center on Neurodegenerative Diseases (CIBERNED), Institute of Health Carlos III, 28029 Madrid, Spain

**Keywords:** copper, dopamine, homovanillic acid, striatum, substantia nigra, Wilson’s disease, microdyalisis, neurotransmission, oxidative stress

## Abstract

Copper is a trace element whose electronic configuration provides it with essential structural and catalytic functions. However, in excess, both its high protein affinity and redox-catalyzing properties can lead to hazardous consequences. In addition to promoting oxidative stress, copper is gaining interest for its effects on neurotransmission through modulation of GABAergic and glutamatergic receptors and interaction with the dopamine reuptake transporter. The aim of the present study was to investigate the effects of copper overexposure on the levels of dopamine, noradrenaline, and serotonin, or their main metabolites in rat’s striatum extracellular fluid. Copper was injected intraperitoneally using our previously developed model, which ensured striatal overconcentration (2 mg CuCl_2_/kg for 30 days). Subsequently, extracellular fluid was collected by microdialysis on days 0, 15, and 30. Dopamine (DA), 3,4-dihydroxyphenylacetic acid (DOPAC), homovanillic acid (HVA), 5-hydroxyindoleacetic acid (5-HIAA), and noradrenaline (NA) levels were then determined by HPLC coupled with electrochemical detection. We observed a significant increase in the basal levels of DA and HVA after 15 days of treatment (310% and 351%), which was maintained after 30 days (358% and 402%), with no significant changes in the concentrations of 5-HIAA, DOPAC, and NA. Copper overload led to a marked increase in synaptic DA concentration, which could contribute to the psychoneurological alterations and the increased oxidative toxicity observed in Wilson’s disease and other copper dysregulation states.

## 1. Introduction

Copper (Cu) is an essential trace element, whose essentiality is mainly derived from its high redox activity, which grants this metal its coordination and catalytic properties. These properties, however, can become detrimental in dyshomeostatic conditions, such as in the hepatoneurological damage present in Wilson’s disease (WD) [1,2,3].

Recently, a new role for Cu as a neurotransmission modulator has attracted significant attention. Cu has been found to be stored in synaptic vesicles, and its release after depolarization produces concentrations of up to 250 µM in the synaptic cleft, exceeding the concentration in cerebrospinal fluid by three orders of magnitude [4]. At low µM concentrations, Cu potentiates NMDA receptor currents [5]. Additionally, Cu can inhibit GABA_A_ receptors and allosterically inhibit the activity of the dopamine transporter (DAT) [6,7,8].

WD courses with psychoneurological manifestations that are still poorly understood [9]. Even heterozygous patients, who do not manifest the disease, exhibit noticeable brain abnormalities [10]. Furthermore, elevated levels of free Cu have been reported in the serum and cerebrospinal fluid of patients with Alzheimer’s disease, suggesting a link between high serum Cu concentrations and neurocognitive decline [11]. These brain abnormalities and psychoneurological impairments associated with Cu dysregulation could be attributed to Cu’s role as a modulator of neurotransmission, in addition to its well-established mechanisms of inducing oxidative stress [1,2].

This study was designed to assess the effects of Cu overexposure on the physiological levels of key neurotransmitters, including dopamine (DA), 3,4-dihydroxyphenylacetic acid (DOPAC), homovanillic acid (HVA), noradrenaline (NA), and 5-hydroxyindoleacetic acid (5-HIAA) in the rat striatum. This was conducted using our established Cu overload model, which has been demonstrated to increase Cu concentrations and induce oxidative stress in the nigrostriatal system [2]—a system potentially implicated in the previously mentioned abnormalities.

## 2. Results

As shown in Figure 1, Cu treatment via intraperitoneal administration (2 mg/kg/day) induced a significant increase in both DA and HVA levels on day 15 (310% and 351%) and day 30 (358% and 402%), while no significant changes were observed in DOPAC, 5-HIAA or NA levels, although a non-significant trend toward increased NA levels was noted. From day 15 to day 30, the increase in both DA and HVA concentrations was maintained but did not show any significant changes. Concentrations for each individual rat (estimated as area under the curve, AUC) at days 15 and 30 were compared with their baseline values on day 0 and expressed as mean percentages ± SEM. The average absolute concentrations of each neurotransmitter at time zero were as follows: DA: 304 fg/mL, HVA: 169 pg/mL, DOPAC: 1862 fg/mL, 5-HIAA: 657 fg/mL, and NA: 314 fg/mL. The absolute neurotransmitter concentration values exhibited high interindividual variability, due to the potential variability in the probe recovery of neurotransmitters and their metabolites in the different microdialysis studies [12]. However, the percentage changes over time were consistent and showed much lower interindividual variability. Therefore, we focused on comparing percentage changes in each individual rat rather than absolute values to account for this variability. All detailed data is available in the Appendix A.

Notably, in a previous study using the same Cu overload model, our group reported that Cu concentrations showed an increase of +59% in the striatum and +68% in substantia nigra pars compacta (SNc) [2].

## 3. Discussion

Axons from the SNc release DA into the synaptic cleft within the striatum. This action is central to the nigrostriatal pathway, which plays crucial roles in movement, reward processing, and cognitive functions. SNc presents 70% of GABAergic inhibitory afferents, while virtually all the rest are glutamatergic, particularly via NMDA receptors [13]. Cu has been shown to effectively inhibit the GABA_A_ receptor with an IC_50_ in the nM range both in vitro and in vivo [6,7,14], whereas at low µM concentrations, it has also been shown to allosterically stimulate NMDA currents in cultured neonatal rat cerebellum granule cells [5]. Both GABAergic antagonism and NMDA agonism lead to SNc excitation, thus triggering burst firing, the most DA-releasing pattern [13]. Additionally, Cu was recently found to be the most effective transition metal in inhibiting DAT affinity for DA in cell cultures [8]. This reduction in DAT activity, combined with the increase in DA release, could contribute to the increase in DA striatal extracellular levels reported here (as illustrated in Figure 2). The absence of significant changes in the concentrations of 5-HIAA and NA rules out the potential influence of these pathways on DA release. The absence of a change in DOPAC suggests an increase in DA in the extracellular space, where COMT can directly convert DA to HVA without the need for intracellular MAO, thus bypassing the DOPAC intermediate.

Using the same Cu overload model, we have previously found evidence of brain oxidative stress [2]. Although the rats in the present study underwent the identical treatment protocol, the cited oxidative stress data pertain to our earlier work in which we characterized this overexposure model. This represents a limitation of the current study, as the specific oxidative stress measurements were not directly obtained from the rats used in the present investigation but from those used during the characterization of the model.

Basal ganglia are especially sensitive to oxidative stress damage produced by transition metals; in fact, Cu overexposure is associated with a higher occurrence of Parkinson’s disease [15]. In WD-associated parkinsonism, there is a preferential death of presynaptic striatal neurons [16], which could be attributed to the potential in situ catalytic activity exerted by Cu on DA autoxidation [2], derived from the supraphysiological concentrations of both substances (DA and Cu) in the striatum. 

In conclusion, Cu overload significantly increased DA and HVA synaptic concentrations, consistent with Cu’s recently described neuromodulatory roles. This dopaminergic upregulation may be crucial in the psychoneurological symptoms observed in WD. Moreover, the deleterious interaction between Cu and DA could contribute to oxidative stress, potentially underlying nigrostriatal cell loss in WD and other Cu dyshomeostatic states.

## 4. Materials and Methods

### 4.1. Rat Treatment

Six adult male Sprague Dawley rats (225–250 g in weight) from the Animal Breeding Unit of the University of Santiago de Compostela were used to perform the studies. Animals were housed in polypropylene cages (two rats per cage; the first few days after surgery rats were housed individually) and maintained at 21 ± 1 °C on a 12 h light/12 h dark cycle with water and food ad libitum. All experiments were carried out in accordance with the European Directive 2010/63/EU and the Spanish RD/53/2013 for the protection of animals used for scientific purposes, and all efforts were made to minimize the number of animals and their suffering. The experimental design was approved by the corresponding committee at the University of Santiago de Compostela. For chronic intraperitoneal (IP) injection and subsequent studies, a total of 6 animals were used. The rats were injected daily with copper (II) chloride (Sigma-Aldrich Co, St. Louis, MO, USA) in saline at a dose of 2 mg CuCl_2_/kg for 30 days. Dosages were adjusted according to the animal’s weight before injection.

### 4.2. Microdialysis Procedure

As reported in previous publications [17], rats were anesthetized and mounted on a stereotaxic frame (Kopf Instruments, Tujunga, CA, USA) in a flat-skull position (tooth bar set at −3.3) for implantation of microdialysis probes in the striatum (0.6 mm anterior to the bregma, −3.5 mm from the midline, and −6.0 mm ventral to the skull). The guide cannula was fixed with two stainless steel screws and dental cement (TriplexCold, Ivoclar Vivadent, Schaan, Liechtenstein). Following surgery, rats were housed individually. The day before the experiment, concentric microdialysis probes (CMA 12 Elite 4 mm, CMA, Kista, Sweden) were introduced, and the rats were placed into a microdialysis system with a balanced arm for freely moving rats (CMA, Kista, Sweden). The probes were perfused with Ringer’s solution and regularly checked to prevent osmolarity stress (140 mM NaCl, 3.0 mM KCl, 1.2 mM CaCl_2_, and 1.0 mM MgCl_2_; Sigma-Aldrich Co, St. Louis, MO, USA) at a constant flow rate of 1 μL/min, and perfusates were collected every 20 min. The probes were rinsed for 50 min, and three baseline samples were then collected prior to L-DOPA injection. Samples were taken over a period of 2 h. Perfused fractions were analyzed by HPLC with electrochemical detection (see bellow) Total levels of neurotransmitters and metabolites were estimated as the area under the curve. 

### 4.3. High-Performance Liquid Chromatography (HPLC)

Levels of DA, HVA, DOPAC, 5-HIAA, and NA were analyzed in the perfusate fractions by high-performance liquid chromatography (HPLC) with electrochemical detection. The perfusates were injected directly into the HPLC equipment (Shimadzu LC prominence, Shimadzu Corporation, Kyoto, Japan). DA, HVA, DOPAC, 5-HIAA, and NA were separated on a Waters Symmetry300C18 analytical column (Waters Corporation, Milford, MA, USA). The mobile phase consisted of a 10% MeOH solution (pH 4) containing 70 mM KH_2_PO_4_, 1 mM octanesulfonic acid, and 1 mM EDTA and was delivered at a rate of 1 mL/min. Detection was performed with a coulometric electrochemical detector (ESA Coulochem III, ESA Biosciences, Chelmsford, MA, USA). The first and second electrodes of the analytical cell were set at +50 mV and +350 mV, respectively, and the guard cell was set at −100 mV. Data were acquired and processed with the Shimadzu LC solution software v1.25 (Shimadzu Corporation, Kyoto, Japan) and expressed as percentages in relation to the corresponding control (time zero) concentration for each rat.

### 4.4. Statistical Analysis

Concentrations were measured at baseline (t = 0), day 15, and day 30, for each individual rat (*n* = 6). The values for each rat were compared with its own values at different time points, as the absolute value is subject to high interindividual variation; hence, we considered percentage changes rather than absolute values. To account for the positive skew in the data, concentrations were normalized to the baseline value and then log-transformed (log10). Normality was verified using the D’Agostino–Pearson and Shapiro–Wilk tests. A repeated-measure ANOVA followed by Tukey’s HSD post hoc test was conducted to identify significant differences (*p* < 0.05) between time points.

## Figures and Tables

**Figure 1 ijms-25-08309-f001:**
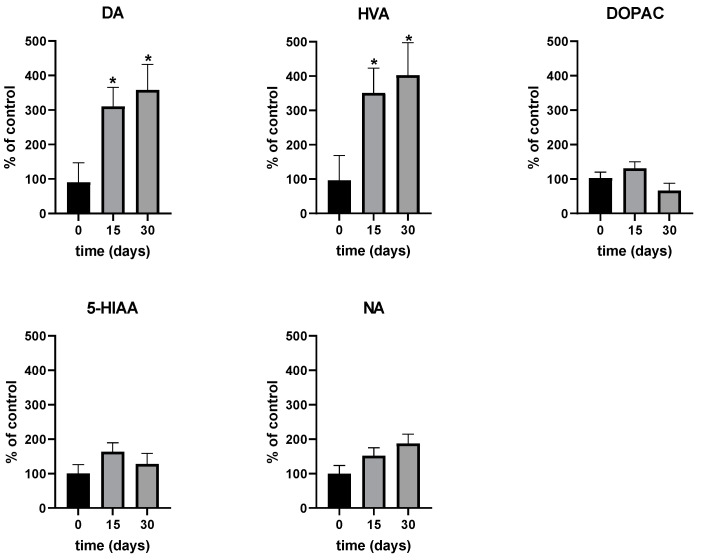
Striatal extracellular levels of dopamine (DA) and its metabolites homovanillic acid (HVA) and 3,4-dihydroxyphenylacetic acid (DOPAC), serotonin metabolite 5-hydroxyindoleacetic acid (5-HIAA), and noradrenaline (NA) determined by HPLC on days 0, 15, and 30, after brain microdialysis in six freely moving rats. Data are expressed as a mean of the percentages ± SEM in relation to the baseline concentration for each rat. * Statistical significance at *p* < 0.05 (one-way ANOVA repeated measures followed by Tukey HSD) was established in comparison with the corresponding baseline value (time zero) for log-transformed concentrations. The average absolute concentrations of each neurotransmitter at time zero were as follows: DA: 304 fg/mL, HVA: 169 pg/mL, DOPAC: 1862 fg/mL, 5-HIAA: 657 fg/mL, and NA: 314 fg/mL. Detailed data is available in the Appendix A.

**Figure 2 ijms-25-08309-f002:**
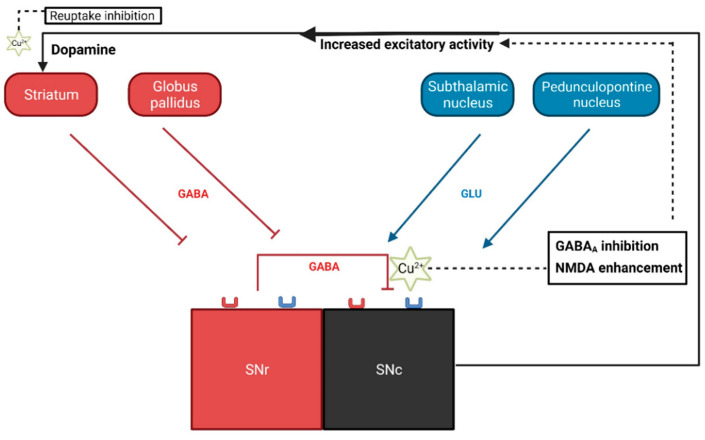
Simplified diagram of the substantia nigra afferent system. Most substantia nigra pars compacta (SNc) afferents (approx. 70%) are GABAergic (inhibited by Copper, Cu), originating from the striatum, globus pallidus, and substantia nigra pars reticulata (SNr), which cause hyperpolarization and consequently a decrease in dopamine (DA) striatal release. Virtually, the rest of SNc afferents are glutamatergic (enhanced by Cu), which induces an increase in DA striatal release. In the nigrostriatal synapse, the released dopamine is reuptaken by the dopamine transporter (DAT, inhibited by Cu).

## Data Availability

Data are contained within the article or Appendix A.

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
