# Peer review of "Copper Overload Increased Rat Striatal Levels of Both Dopamine and Its Main Metabolite Homovanillic Acid in Extracellular Fluid"

_ijms, 2024, doi:10.3390/ijms25158309_

Round 1

Reviewer 1 Report

Comments and Suggestions for Authors

Overall this is a well-written manuscript.   The authors need to add the number rats used in each group to the manuscript.  There is no statistical analysis section in methods.  Although statistical analysis methods are mentioned in the legend of figure 1 this information should be added in greater detail to the methods section under an appropriate subheading.     Data are all presented as percent change. Adding the actual values for significant increases to the manuscript either within the text or in the figure legends would be helpful to readers so they do not have to search through supplemental tables for this information.   There is no discussion of the limitations of this research.  Please add a discussion of the limitations of this research.  Correlating data with a biomarker of oxidative stress in these same animals (if this data is available) would strengthen this paper; if oxidative stress biomarker data is not available then this is a limitation of this work which should be discussed.

Author Response

1)The authors need to add the number rats used in each group to the manuscript.  

Thank you for pointing this out. We of course agree the number of rats was lacking. It has been added in figure legend (line 66) and Materials and Methods section (line 132, 174)

2)There is no statistical analysis section in methods.  Although statistical analysis methods are mentioned in the legend of figure 1 this information should be added in greater detail to the methods section under an appropriate subheading.    

Absolutely correct. It has been added to Materials and methods section, lines 173-181.

3) Data are all presented as percent change. Adding the actual values for significant increases to the manuscript either within the text or in the figure legends would be helpful to readers so they do not have to search through supplemental tables for this information.   

Added both in Results section and in figure legends (line 69-71) and Results (lines 79-86, together with the explanation of why we work with percentage of change) 

4)There is no discussion of the limitations of this research.  Please add a discussion of the limitations of this research.  Correlating data with a biomarker of oxidative stress in these same animals (if this data is available) would strengthen this paper; if oxidative stress biomarker data is not available then this is a limitation of this work which should be discussed.

Addition of an explanation of this limitation in lines 108-112

Reviewer 2 Report

Comments and Suggestions for Authors

The study by Cruces-Sande et al examined the changes in dopamine and its metabolites following prolonged exposure to copper for 30 days. The communication paper is of interest and addresses an valuable point regarding external factors and their contribution to neurodegenerative disorders.

However, there are some points that needs to be clarified before publication.

1) Since HPLC is a quantitative method, the levels of measured parameters should be expressed in absolute numbers. This is especially valuable to other researchers since it could increase the reproducibility.

2) Please add individual values for each animal in the graphs and since you obtained each value from individual animal, data should be presented with SD.

3) Are you certain that the flow rate was 1ml/min? The outlet volume of your probe is 0.3 ul. It is not likely to have such high flow rate?

4) Have you checked for osmolarity of your Ringer solution to exclude any osmotic stress?

5) It is not clear, have you collected all timepoints from the same animal? I assume so given that you used RM-ANOVA?

Author Response

  1. Since HPLC is a quantitative method, the levels of measured parameters should be expressed in absolute numbers. This is especially valuable to other researchers since it could increase the reproducibility.

Thank you for pointing this out. Absolute values of what was considered 100% have been added, both in the figure legend (lines 69-71) and in the results section (lines 79-86), along with the reasoning which made it preferable for us to work with percentage changes instead of concentration values.

  1. Please add individual values for each animal in the graphs and since you obtained each value from individual animal, data should be presented with SD.

I am not sure if this comment is already addressed by the modifications made in response to the previous comment. The individual values for each rat can be found in the datasheet. However, if it is necessary to add a table to the manuscript with every value for each of the six rats at every time point, I would be happy to do so.

  1. Are you certain that the flow rate was 1 ml/min? The outlet volume of your probe is 0.3 ul. It is not likely to have such a high flow rate?

Thank you for pointing this out, it was actually an error. The flow rate was 1 uL/min. This has been corrected in the manuscript (lines 155-156).

  1. Have you checked for osmolarity of your Ringer solution to exclude any osmotic stress?

Yes, this information has been added in line 155.

  1. It is not clear, have you collected all timepoints from the same animal? I assume so given that you used RM-ANOVA?

That is correct. We compared each of the 6 animals with itself at different time points. This is now more thoroughly explained in lines 79-83 and 173-181.